# Random Walk Diffusion for Efficient Large-Scale Graph Generation

**Tobias Bernecker**[*]  *tobias.bernecker@helmholtz-munich.de*
*Computational Health Center, Helmholtz Munich*
*School of Computation, Information and Technology, Technical University of Munich*

**Ghalia Rehawi**[*]  *ghalia.rehawi@helmholtz-munich.de*
*Computational Health Center, Helmholtz Munich*
*Max Planck Institute of Psychiatry, Munich*
*School of Life Sciences, Technical University of Munich*

**Francesco Paolo Casale**  *francescopaolo.casale@helmholtz-munich.de*
*Computational Health Center, Helmholtz Munich*
*Helmholtz Pioneer Campus, Helmholtz Munich*
*School of Computation, Information and Technology, Technical University of Munich*

**Janine Knauer-Arloth**  *arloth@psych.mpg.de*
*Max Planck Institute of Psychiatry, Munich*
*Computational Health Center, Helmholtz Munich*

**Annalisa Marsico**  *annalisa.marsico@helmholtz-munich.de*
*Computational Health Center, Helmholtz Munich*

**Reviewed on OpenReview:** *https://openreview.net/forum?id=tSFpsfndE7&noteId=bXuL6ssguM*

## Abstract

Graph generation addresses the problem of generating new graphs that have a data distribution similar to real-world graphs. While previous diffusion-based graph generation methods have shown promising results, they often struggle to scale to large graphs. In this work, we propose ARROW-Diff (AutoRegressive RandOm Walk Diffusion), a novel random walk-based diffusion approach for efficient large-scale graph generation. Our method encompasses two components in an iterative process of random walk sampling and graph pruning. We demonstrate that ARROW-Diff can scale to large graphs efficiently, surpassing other baseline methods in terms of both generation time and multiple graph statistics, reflecting the high quality of the generated graphs.

## 1 Introduction

Graph generation addresses the problem of generating graph structures with specific properties similar to real-world ones, with applications ranging from modeling social interactions to constructing knowledge graphs, as well as designing new molecular structures. For example, an emerging task in drug discovery is to generate molecules with high drug-like properties. Traditional methods for graph generation focused on generating graphs with a predefined characteristic (Erdős et al., 1960; Barabási & Albert, 1999). Due to their hand-crafted nature, these methods fail to capture other graph properties, e.g., the graphs generated by Barabási & Albert (1999) are designed to capture the scale-free topology but fail to capture community structures of real-world graphs.

---

[*]Equal contribution.

Recently, deep learning-based approaches have gained attention for overcoming the limitations of traditional graph generation methods by learning the complex topology of real-world graphs. These approaches typically involve an encoder that learns a dense representation of the graph, a sampler that generates samples from this representation, and a decoder which restores the sampled representation into a graph structure (Zhu et al., 2022). Different decoding strategies have been proposed, including sequential decoders that generate the graph step-by-step (You et al., 2018; Liao et al., 2019), and one-shot generators that can create the entire graph in a single step (Kipf & Welling, 2016b; Simonovsky & Komodakis, 2018; Grover et al., 2019). While sequential decoders struggle to model long-term dependencies and require a node-ordering process, one-shot generators suffer from scalability issues and compromised quality due to the independent treatment of edges (Chanpuriya et al., 2021), making these methods suboptimal for large-scale graph generation (Guo & Zhao, 2022).

An even more recent body of work in graph generation are diffusion-based probabilistic models, inspired by non-equilibrium thermodynamics and first introduced by Sohl-Dickstein et al. (2015). Briefly, diffusion models consist of two processes: A forward diffusion process which gradually corrupts input data until it reaches pure noise, and a reverse diffusion process which learns the generative mechanism of the original input data using a neural network. Diffusion-based methods for graph generation can be divided into two main categories. The first one includes methods that implement diffusion in the continuous space e.g., by adding Gaussian noise to the node features and graph adjacency matrix (Niu et al., 2020; Jo et al., 2022). This form of diffusion however makes it difficult to capture the underlying structure of graphs since it destroys their sparsity pattern (Vignac et al., 2023). The second category includes methods that are based on diffusion in the discrete space (Haefeli et al., 2022; Vignac et al., 2023; Chen et al., 2023) by performing successive graph modifications e.g., adding or deleting edges/nodes or edge/node features. Diffusion-based graph generation methods do not suffer from long-term memory dependency which makes them advantageous over autoregressive (sequential) methods. However, many approaches found in the literature are only designed for small graphs (Niu et al., 2020; Jo et al., 2022; Vignac et al., 2023). One of the recent diffusion-based graph generation approaches is EDGE (Chen et al., 2023), which uses a combination of a discrete diffusion model and a Graph Neural Network (GNN). EDGE scales to large graphs (up to ∼4k nodes and ∼38k edges), however, the number of required diffusion steps and hence graph generation time increases linearly with the number of edges in the graph.

To address the existing gaps in this field, we introduce ARROW-Diff (AutoRegressive RandOm Walk Diffusion), a novel iterative procedure designed for efficient, high-quality, and large-scale graph generation. To the best of our knowledge, ARROW-Diff is the first method to perform (discrete) diffusion on the level of random walks within a graph. Our method overcomes many limitations of existing graph generation approaches, including scalability issues, the independent treatment of edges, the need for post-processing, and the high computational complexity resulting from operations performed directly on the adjacency matrix. ARROW-Diff integrates two components: (1) A discrete diffusion model based on the order-agnostic autoregressive diffusion framework (OA-ARDM) (Hoogeboom et al., 2022) applied on random walks sampled from a real-world input graph, and (2) a GNN trained to predict the validity of the proposed edges in the generated random walks from component (1). The random walk-based diffusion in component (1) enables ARROW-Diff to capture the complex and sparse structure of graphs (Perozzi et al., 2014; Grover & Leskovec, 2016; Bojchevski et al., 2018), avoid any dense computation, and scale to large graphs, since the required number of diffusion steps is only equal to the random walk length. Moreover, due to the autoregressive nature of the implemented diffusion framework (OA-ARDM), ARROW-Diff overcomes the edge-independence limitation of methods like VGAE (Kipf & Welling, 2016b) and NetGAN (Bojchevski et al., 2018). On the other hand, the GNN in component (2) enables ARROW-Diff to select highly probable edges based on the topological structure learned from the original graph. It also helps to overcome the need for post-processing as in NetGAN (Bojchevski et al., 2018). ARROW-Diff builds the final graph by integrating the two components in an iterative process. This process is further guided by node degrees inspired by EDGE (Chen et al., 2023) to generate graphs with a similar degree distribution to the original graph. Using this iterative approach, we show that ARROW-Diff demonstrates superior or comparable performance to other baselines on multiple graph statistics. Furthermore, ARROW-Diff is able to efficiently generate large graphs (in this work, up to ∼20k nodes and ∼60k edges), such as the citation networks from McCallum et al. (2000); Sen

et al. (2008); Pan et al. (2016), with significantly reduced (down to almost 50%) generation time compared to other baselines.

## 2  Related Work

In the following, we provide an overview of the different classes of graph generation models and identify their strengths, as well as their limitations.

**One-Shot Graph Generation Models**   These methods include graph generation approaches based on the Variational Autoencoder (VAE) (Kingma & Welling, 2013) model, like VGAE (Kipf & Welling, 2016b), GraphVAE (Simonovsky & Komodakis, 2018) and Graphite (Grover et al., 2019), which embed a graph $G$ into a continuous latent representation $z$ using an encoder defined by a variational posterior $q_\phi(z|G)$, and a generative decoder $p_\theta(G|z)$. These models are trained by minimizing the upper bound on the negative log-likelihood $\mathbb{E}_{q_\phi(z|G)}[-\log p_\theta(G|z)] + \mathrm{KL}[q_\phi(z|G)\,\|\,p(z)]$ (Kingma & Welling, 2013). However, due to their run time complexity of $\mathcal{O}(N^2)$, VAE-based graph generation approaches are unable to scale to large graphs.

**Sequential Graph Generation Models**   One of the most scalable non-diffusion and sequential graph generation methods is NetGAN (Bojchevski et al., 2018), which is based on the concept of Generative Adversarial Networks (GANs) (Goodfellow et al., 2014). Specifically, it uses a Long Short-Term Memory (LSTM) (Hochreiter & Schmidhuber, 1997) network to generate random walks. After training, the generated random walks are used to construct a score matrix from which the edges of the generated graph are sampled. The aforementioned approach generates edges in an independent manner, sacrificing the quality of the generated graphs and limiting their ability to reproduce some statistics of the original graphs such as triangle counts and clustering coefficient (Chanpuriya et al., 2021). In contrast, ARROW-Diff overcomes this problem by directly using the edges generated by an autoregressive diffusion process on the level of random walks. Other edge-dependent sequential approaches include GraphRNN (You et al., 2018), GRAN (Liao et al., 2019), and BiGG (Dai et al., 2020). These methods iteratively generate the entries of a graph adjacency matrix one entry or one block of entries at a time. To overcome the long-term bottleneck issue of Recurrent Neural Networks (RNNs), Liao et al. (2019) propose to use a GNN architecture instead of an RNN, which utilizes the already generated graph structure to generate the next block, allowing it to model complex dependencies between each generation step. To satisfy the permutation invariance property of graphs, these methods require a node ordering scheme. Moreover, GRAN and GraphRNN are only able to scale to graphs of up to 5k nodes.

**Discrete Diffusion-Based Graph Generation Models**   To exploit the sparsity property of graphs, discrete diffusion-based graph generation models focus on diffusion in the discrete space, i.e., on the level of the adjacency matrix (Haefeli et al., 2022; Vignac et al., 2023). Although these approaches generate high-quality graphs (Niu et al., 2020; Jo et al., 2022) and overcome the limitation of autoregressive models, they are restricted to small graph generation, like chemical molecules, because they perform predictions for every pair of nodes. Recently, Kong et al. (2023) developed GraphARM, an autoregressive graph generation approach based on diffusion in the node space. GraphARM demonstrates faster sampling time, since the number of diffusion steps required equals the number of nodes. Our method, however, requires only a number of diffusion steps equal to the random walk length, making it far more scalable than GraphARM. Aside from the method proposed in this work, the only diffusion-based approach able to scale to large graphs is EDGE (Chen et al., 2023). The forward diffusion process of EDGE is defined by successive edge removal until an empty graph is obtained. In the reverse diffusion process, only a fraction of edges are predicted between active nodes for which the degree changes during the forward diffusion process. This method generates graphs with a similar degree distribution to the original graph and has a decreased run time of $\mathcal{O}(T \max(M, K^2))$, where $M$ is the number of edges in a graph and $K$ is the number of active nodes. In this work, we introduce a random walk-based discrete diffusion approach to efficiently generate random walks as a first step for graph generation. The random walk-based diffusion process implemented through the OA-ARDM (Hoogeboom et al., 2022) framework enables more efficient sampling than in other discrete diffusion models, with significantly reduced number of diffusion steps equal only to the random walk length.

Additionally, a GNN component validates edges from the sampled random walks, enabling us to refine the generated graph and produce graphs of high quality.

## 3 Method

In this section, we introduce ARROW-Diff, an iterative procedure for efficient large-scale graph generation. We designed ARROW-Diff in a way to overcome the critical limitations of previous graph generation approaches and to enable it to scale to large graphs efficiently, without sacrificing the graph generation quality. Figure 1 provides an overview of ARROW-Diff, which leverages two models that are trained independently of each other on a training set of edges from the original graph: (1) A random walk-based diffusion model based on the OA-ARDM (Hoogeboom et al., 2022) framework trained to generate random walks from the original graph. This involves first sampling of random walks and then masking nodes in the forward diffusion process, with the model predicting the original nodes in the reverse diffusion process. (2) A GNN trained to capture the topology of the original graph. Our proposed method, ARROW-Diff, incorporates these two components in an iterative process in which (1) the diffusion model generates random walks, proposing potential edges, and (2) the GNN evaluates and filters the edges proposed by the OA-ARDM based on learned topological features. This process is further guided by changes in the node degrees to generate the final graph. In the following, we provide background information on the OA-ARDM and its adaptation for random walk-based diffusion, followed by a detailed description of ARROW-Diff.

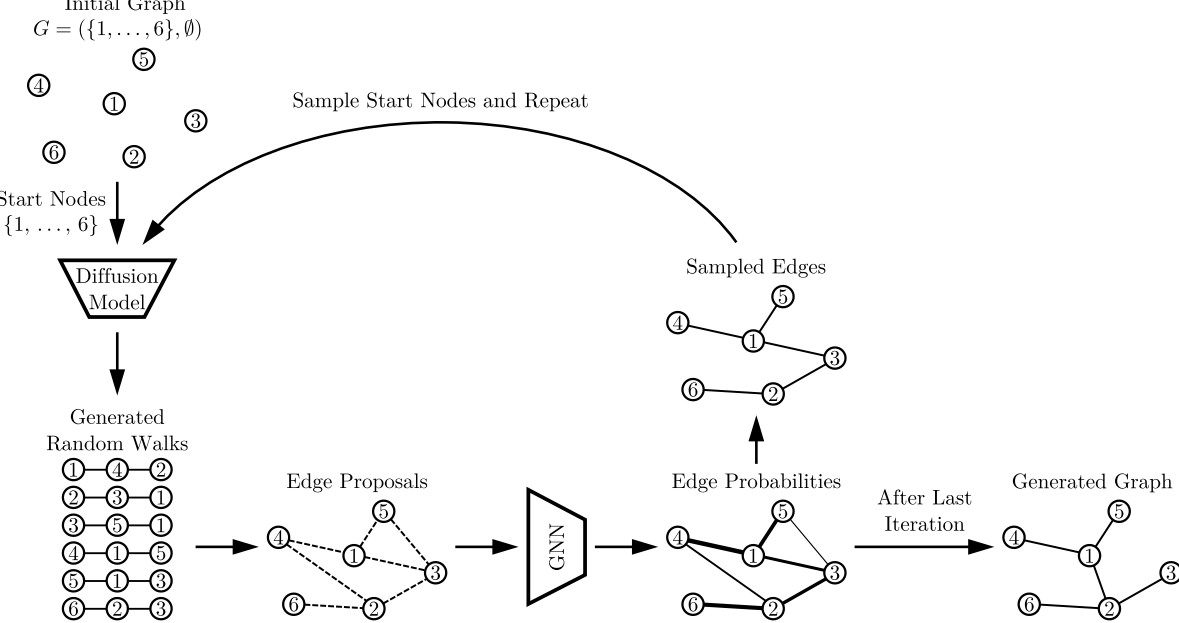

Figure 1: Overview of ARROW-Diff graph generation (inference) using a trained OA-ARDM (Hoogeboom et al., 2022) and a trained GNN. Iteratively, and starting from an empty graph, a diffusion model samples random walks from a set of start nodes. Then, a GNN classifies the proposed edges and filters out invalid ones. This procedure is repeated $L$ times using a different set of sampled start nodes guided by the change of node degrees with respect to the original graph.

**Background: Order Agnostic Autoregressive Diffusion Models** Recent works show that diffusion models are applicable to discrete data (Sohl-Dickstein et al., 2015; Hoogeboom et al., 2021; Austin et al., 2021; Hoogeboom et al., 2022). The diffusion process of these models is based on the Categorical distribution over input features of a data point, instead of the Gaussian distribution. Initially, discrete diffusion models used uniform noise to corrupt the input in the forward diffusion process (Sohl-Dickstein et al., 2015; Hoogeboom

---

**Algorithm 1** Optimizing Random Walk OA-ARDMs

---

1: **Input:** A random walk $\boldsymbol{x} \in V^D$, the number of nodes $N = |V|$, and a network $f$.
2: **Output:** ELBO $\mathcal{L}$.
3: Sample $t \sim \mathcal{U}(1, \dots, D)$
4: Sample $\sigma \sim \mathcal{U}(S_D)$
5: Compute $\boldsymbol{m} \leftarrow (\sigma < t)$
6: Compute $\boldsymbol{i} \leftarrow \boldsymbol{m} \odot \boldsymbol{x} + (1 - \boldsymbol{m}) \odot ((N+1) \cdot \boldsymbol{1}_D)$
7: $\boldsymbol{l} \leftarrow (1 - \boldsymbol{m}) \odot \log \mathcal{C}(\boldsymbol{x}|f(\boldsymbol{i}, t))$
8: $\mathcal{L}_t \leftarrow \frac{1}{D-t+1} sum(\boldsymbol{l})$
9: $\mathcal{L} \leftarrow D \cdot \mathcal{L}_t$

---

et al., 2021). Later, Austin et al. (2021) extended this process and introduced a general framework for discrete diffusion (D3PM) based on Markov transition matrices $[\boldsymbol{Q}_t]_{ij} = q(x_t = j|x_{t-1} = i)$ for categorical random variables $x_{t-1}, x_t \in \{1, 2, \dots, K\}$. One possible realization of the D3PM framework is the so-called absorbing state diffusion (Austin et al., 2021) that uses transition matrices with an additional absorbing state to stochastically mask entries of data points in each forward diffusion step. Recently, Hoogeboom et al. (2022) introduced the concept of OA-ARDMs combining order-agnostic autoregressive models (Uria et al., 2014) and absorbing state diffusion. Unlike standard autoregressive models, order-agnostic autoregressive models are able to capture dependencies in the input regardless of their temporal order. Let $\boldsymbol{x}$ be a $D$-dimensional data point. An order-agnostic autoregressive model can generate $\boldsymbol{x}$ in a random order that follows a permutation $\sigma \in S_D$, where $S_D$ denotes the set of possible permutations of $\{1, 2, \dots, D\}$. Specifically, their log-likelihood can be written as

$$\log p(\boldsymbol{x}) \geq \mathbb{E}_{\sigma \sim \mathcal{U}(S_D)} \sum_{t=1}^{D} \log p(x_{\sigma(t)}|\boldsymbol{x}_{\sigma(<t)}), \tag{1}$$

where $\boldsymbol{x}_{\sigma(<t)} = \{x_i | \sigma(i) < t, i \in \{1, \dots, D\}\}$ represents all elements of $\boldsymbol{x}$ for which $\sigma(i)$ is less than $t$ (Hoogeboom et al., 2022). In the following, we explain the proposed random walk-based autoregressive diffusion in more detail.

**Random Walk Diffusion** Consider a graph $G = (V, E)$ with $N = |V|$ nodes. We aim to learn the (unknown) generative process $p(G)$ of $G$. Inspired by DeepWalk (Perozzi et al., 2014), node2vec (Grover & Leskovec, 2016), and by the random walk-based graph generation approach introduced by Bojchevski et al. (2018), we suggest to sample random walks from a trained diffusion model and use the edges comprising the walks as proposals for generating a new graph. To achieve this, we train an OA-ARDM (Hoogeboom et al., 2022) by viewing each node in a random walk as a word in a sentence, and follow the proposed training procedure of Hoogeboom et al. (2022) for OA-ARDMs on sequence data (Algorithm 1): For a random walk $\boldsymbol{x} \in V^D$ of length $D$, we start by sampling a time step $t \sim \mathcal{U}(1, \dots, D)$ and a random ordering of the nodes in the walk, $\sigma \sim \mathcal{U}(S_D)$, both from a uniform distribution. For each time step $t$ of the diffusion process, a BERT-like (Devlin et al., 2018) training is performed, in which $D - t + 1$ nodes (words) are masked and then predicted by a neural network. Specifically, the OA-ARDM is trained by maximizing the following log-likelihood component at each time step $t$ (Hoogeboom et al., 2022):

$$\mathcal{L}_t = \frac{1}{D-t+1} \mathbb{E}_{\sigma \sim \mathcal{U}(S_D)} \sum_{k \in \sigma(\geq t)} \log p(x_k|\boldsymbol{x}_{\sigma(<t)}) \tag{2}$$

In the case of diffusion on random walks, masking of nodes is equivalent to setting them to an absorbing state represented by an additional class $N+1$ (Hoogeboom et al., 2022). Thus, as suggested by Hoogeboom et al. (2022), the inputs to the network are (1) the masked random walk $\boldsymbol{i} = \boldsymbol{m} \odot \boldsymbol{x} + (1 - \boldsymbol{m}) \odot \boldsymbol{a}$, where $\boldsymbol{m} = \sigma < t$ is a Boolean mask, $\boldsymbol{a} = (N+1) \cdot \boldsymbol{1}_D$ and $\boldsymbol{1}_D$ is a $D$-dimensional vector of ones, and (2) the sampled time step $t$. During the training of the OA-ARDM, the random walks are sampled from the original graph.

**ARROW-Diff Graph Generation** Our proposed graph generation method, ARROW-Diff, is outlined in Algorithm 2. ARROW-Diff generates new graphs similar to a single, given original graph $G = (V, E)$ through

---

**Algorithm 2** ARROW-Diff Graph Generation

---

1: **Input:** A trained OA-ARDM, a trained GNN. The node set $V$, features $\boldsymbol{X}$ and degrees $\boldsymbol{d}_G$ of an original graph $G$ with the same node ordering as for training the OA-ARDM. The number of steps $L$ to generate the graph.
2: **Output:** A generated graph $\hat{G} = (V, \hat{E})$.
3: Start with an empty graph $\hat{G} = (V, \hat{E})$, where $\hat{E} = \emptyset$
4: Set the initial start nodes $V_{\text{start}}$ to all nodes in the graph: $V_{\text{start}} = V$
5: **for** $l = 1$ **to** $L$ **do**
6:     Sample one random walk for each start node $n \in V_{\text{start}}$ using the OA-ARDM: $\mathcal{R}$
7:     Compute edge proposals from $\mathcal{R}$:
     $\hat{E}_{\text{proposals}} := \{(n_i, n_j) \in \mathcal{R} | n_i, n_j \in V, i \neq j\}$
8:     Run the GNN on $G = (V, \hat{E} \cup \hat{E}_{\text{proposals}}, \boldsymbol{X})$ to obtain probabilities for all edges in $\hat{E} \cup \hat{E}_{\text{proposals}}$
9:     Sample valid edges $\hat{E}_{\text{valid}}$ from $\hat{E} \cup \hat{E}_{\text{proposals}}$ according to the edge probabilities
10:     Edge update: $\hat{E} \leftarrow \hat{E}_{\text{valid}}$
11:     **if** $l < L$ **then**
12:         Compute the node degrees $\boldsymbol{d}_{\hat{G}}$ of $\hat{G}$ based on $\hat{E}$
13:         Compute $\boldsymbol{d} := \max(0, \boldsymbol{d}_G - \boldsymbol{d}_{\hat{G}})$
14:         Compute a probability for each node $n \in V$: $p(n) = \frac{d_n}{\max(\boldsymbol{d})}$
15:         Sample start nodes $V_{\text{start}}$ from $V$ according to $p(n)$ using a Bernoulli distribution
16:     **end if**
17: **end for**

---

an iterative procedure of $L$ iterations: It starts with an empty graph $\hat{G} = (V, \hat{E} = \emptyset)$, which contains the same set of nodes $V$ as the original graph but no edges. To add edges to $\hat{G}$, ARROW-Diff initially samples one random walk for every node in $V$ and considers all edges in these random walks as edge proposals $\hat{E}_{\text{proposals}}$ for $\hat{G}$ (lines 6 and 7 in Algorithm 2). Next, and similar to the work of Liao et al. (2019), we sample valid edges from the proposed ones $\hat{E}_{\text{proposals}}$ using a Bernoulli distribution based on the binary edge classification probabilities. These probabilities are calculated by taking the sigmoid of the dot-product of node embeddings (for node-pairs inside $\hat{E} \cup \hat{E}_{\text{proposals}}$) predicted by the GNN component on $G = (V, \hat{E} \cup \hat{E}_{\text{proposals}}, \boldsymbol{X})$ (line 8). This GNN model is trained on a perturbed version of a training set of edges from the original graph $G$ to perform edge classification. Specifically, the training set of edges is corrupted by deleting edges and inserting invalid (fake) edges. Inspired by the degree-guided graph generation process of Chen et al. (2023), we now sample nodes from $V$ using a Bernoulli distribution by considering each node $n \in V$ according to a success probability $p(n) = \frac{d_n}{\max(\boldsymbol{d})}$, where $\boldsymbol{d} := \max(0, \boldsymbol{d}_G - \boldsymbol{d}_{\hat{G}})_n$ are the positive differences of node degrees $\boldsymbol{d}_G$ and $\boldsymbol{d}_{\hat{G}}$ from $G$ and $\hat{G}$. This set of sampled nodes is then used as start nodes for sampling the random walks in the next iteration. Hence, we modify the sampling procedure of Hoogeboom et al. (2022), which originally starts from a sequence with only masked tokens, by manually setting the first node of a random walk $\boldsymbol{x}$ to a specific node $n \in V$, i.e.

$$x_k = \begin{cases} n & \text{if } k = 1, \\ \text{mask} & \text{if } k \in \{2, \dots, D\}. \end{cases} \tag{3}$$

Additionally, we use a restricted set of permutations $S_D^{(1)} := \{\sigma \in S_D | \sigma(1) = 1\}$, in which the order of the first element does not change after applying the permutation. To sample the remaining parts $\boldsymbol{x}_{2:D}$ of the random walk $\boldsymbol{x}$, we follow the sampling procedure of Hoogeboom et al. (2022) by starting at time step $t = 2$ and using $\sigma \sim \mathcal{U}(S_D^{(1)})$. ARROW-Diff can generate directed and undirected graphs. To generate undirected graphs, we suggest adding all reverse edges $\{(n_j, n_i) | (n_i, n_j) \in \hat{E}_{\text{proposals}}\}$ to the edge proposals $\hat{E}_{\text{proposals}}$ (line 7), and to sample edges from $\hat{E} \cup \hat{E}_{\text{proposals}}$ in a way to obtain undirected edges in $\hat{E}_{\text{valid}}$ (line 9).

# 4   Experiments and Results

In our experiments, we compare ARROW-Diff to six baseline methods for graph generation that can scale to large graphs. These methods cover the different graph generation techniques which are explained in Section 1 and Section 2. To evaluate the performance of all methods, we use five different real-world citation graph datasets, each containing a single undirected graph and all considered large graphs with varying numbers of nodes/edges. We also evaluate the performance of ARROW-Diff in generating more structured and non-scale-free graphs. To this end, we create a synthetic Stochastic Block Model (SBM) (Holland et al., 1983) graph with 360 nodes and 3,824 edges. The results on the SBM graph are shown in Appendix Section C.

## 4.1   Experimental Setup

**Datasets**   To evaluate ARROW-Diff, we use five citation graph datasets: Cora-ML (McCallum et al., 2000), Cora (McCallum et al., 2000), CiteSeer (Giles et al., 1998), DBLP (Pan et al., 2016), and PubMed (Sen et al., 2008). For Cora-ML and Cora, we use the pre-processed versions from Bojchevski & Günnemann (2018). Each of the five datasets contains one single, undirected, large-scale citation graph. Motivated by Bojchevski et al. (2018), we only take the largest connected component (LCC) of Cora-ML, Cora, CiteSeer, and DBLP, which contain multiple connected components. Table 1 provides an overview of different characteristics for each graph/LCC. Similar to Bojchevski et al. (2018), we split the edges of each graph into training, validation, and test parts, and use only the training edges to train the baseline methods, as well as the OA-ARDM and GNN for ARROW-Diff.

Table 1: Dataset statistics of single, large-scale graph datasets used in this paper: Number of nodes, number of undirected edges, number of node features, and average node degree. A star ($\star$) indicates that the statistics are reported for the LCC of the respective dataset.

| Dataset | # Nodes | # Edges | # Node Features | Avg. Degree |
|---|---|---|---|---|
| Cora-ML$^\star$ | 2,810 | 7,981 | 2,879 | 5.7 |
| Cora$^\star$ | 18,800 | 62,685 | 8,710 | 6.7 |
| CiteSeer$^\star$ | 1,681 | 2,902 | 602 | 3.5 |
| DBLP$^\star$ | 16,191 | 51,913 | 1,639 | 6.4 |
| PubMed | 19,717 | 44,324 | 500 | 4.5 |

**ARROW-Diff Training**   In the following, we provide details about the training of the OA-ARDM (Hoogeboom et al., 2022) and the GNN used in ARROW-Diff, which are trained independently. We train the OA-ARDM for random walk diffusion following the work of Hoogeboom et al. (2022), which is explained in Section 3. Specifically, we use a U-Net (Ronneberger et al., 2015) architecture similar to Ho et al. (2020) with one ResNet block and two levels for the down- and up-sampling processes. Similar to Bojchevski et al. (2018) we set the random walk length, which equals the number of diffusion steps, to $D = 16$ and sample batches of random walks from the training split comprising edges from the original graph. Each node in the random walks, as well as each diffusion time step, is then represented using 64-dimensional learnable embeddings and passed as input to the U-Net. For the GNN component, we train a two-layer GCN (Kipf & Welling, 2016a) to predict the validity of edges based on perturbed versions of the training split of the input graph. Specifically, the GCN is trained by minimizing the binary cross-entropy loss between predicted edge probabilities and the ground truth labels which indicate whether an edge in the perturbed graph is valid or invalid. The predicted probabilities are computed by taking the sigmoid of the dot-product between pairs of node embeddings. We ensure that the GCN model is well-tuned through: (1) The use of a perturbed version of the original graph for the training of the GCN, and (2) the use of a validation split and early stopping on the validation loss to ensure adequate training. As input to the GCN, we use either the original node features (ARROW-Diff with node features) or positional encodings (Vaswani et al., 2017) (ARROW-Diff without node features). The positional encodings are 64-dimensional for Cora-ML and CiteSeer, and 128-dimensional for Cora, DBLP, and PubMed. The GCN uses node embeddings of sizes 100 and 10 in the

hidden and output layer, respectively. The full list of hyper-parameters for training the OA-ARDMs and the GNN models can be found in our code repository `https://github.com/marsico-lab/arrow-diff`.

**Baseline Methods** To compare against ARROW-Diff, we use six different graph generation baseline methods, which are suitable for large graph generation and can be trained on single-graph datasets: VGAE (Kipf & Welling, 2016b), Graphite (Grover et al., 2019), NetGAN Bojchevski et al. (2018), GraphRNN (You et al., 2018), BiGG (Dai et al., 2020), and EDGE (Chen et al., 2023). Hence, previously mentioned methods like DiGress (Vignac et al., 2023) and GDSS (Jo et al., 2022), which are suitable for small graph generation, are excluded from our experiments. GraphRNN can scale to graphs with up to 5k nodes (You et al., 2018), so we only run it on the two smaller datasets Cora-ML (McCallum et al., 2000) and CiteSeer (Giles et al., 1998), since it is still affordable to train it on graphs of such sizes. The recent autoregressive diffusion-based graph generation method GraphARM (Kong et al., 2023) is also excluded due to the absence of a code repository. However, as demonstrated in the complexity analysis in Section 5, ARROW-Diff outperforms GraphARM in terms of runtime. To train the baseline methods, we use the recommended hyper-parameters from their papers and code. The training of NetGAN is performed using their proposed VAL-criterion (Bojchevski et al., 2018) for early stopping on the validation edges from the data split. Some models for EDGE were not showing convergence even after 4 days of training. Hence, for evaluation, we considered the models at epochs 2600 (CiteSeer), 5550 (Cora-ML), 250 (Cora), 450 (DBLP), and 250 (PubMed). Additionally, to fit into GPU memory, we decreased the batch size from 4 to 2 to train on the Cora, DBLP, and PubMed datasets. We also decreased the embedding dimension parameter for BiGG from 256 to 128 for PubMed and to 64 for Cora and DBLP, to fit into memory of a single GPU. Some of the aforementioned baselines do not incorporate node features in their training, hence, for a fair comparison, we report the performance of ARROW-Diff both with and without the use of original node features. In our case, node features are used for training and inference of the GNN model.

**Evaluation of Generated Graphs** We use six different graph metrics to evaluate the performance of the trained models. Additionally, we report the edge overlap (EO) between the generated graphs and the original graph/LCC. Specifically, we generate 10 graphs per dataset and compute the mean and standard deviation of the metrics to obtain a more accurate estimate of the performance and assess robustness.

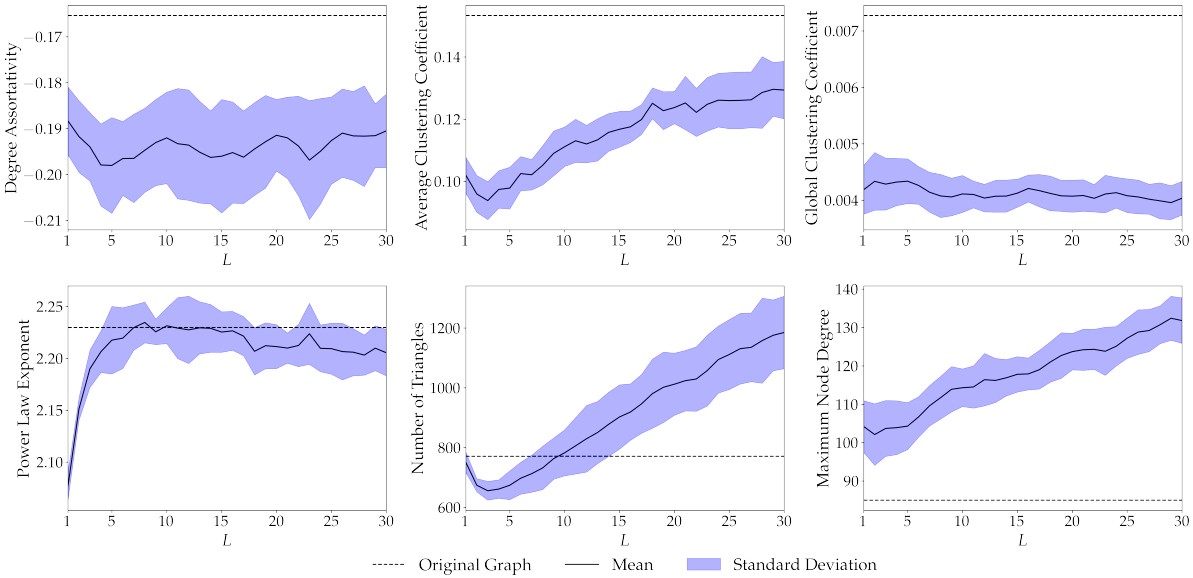

Figure 2: The change in different graph evaluation metrics with respect to $L$. The dotted line reports the value corresponding to the ground truth graph, in this case CiteSeer. For each metric, the mean and standard deviation were computed across 10 generated graphs using $L \in [1, 30]$ iterations for ARROW-Diff.

## 4.2 Hyper-parameter Tuning

ARROW-Diff generates a graph within several iterations $L$. In the following, we investigate the influence of the choice of $L$ on the performance of ARROW-Diff. In Figure 2 we show how the different graph evaluation metrics change with respect to $L$ on the CiteSeer (Giles et al., 1998) dataset. The increase trend on the average clustering coefficient, number of triangles, and maximum node degree can be explained by the increase in the total number of edges of the generated graph with respect to $L$ (see Figure 3). With increasing $L$, more edges are added, which in turn increase the maximum degrees of some nodes as well as the number of triangles. This also explains why e.g. the global clustering coefficient remains mostly constant, since the number of closed and open triangles rise simultaneously. We choose $L = 10$ to generate graphs with ARROW-Diff across all datasets, since this value shows the closest results to the ground truth graph both in terms of number of triangles and power-law exponent. In Figure 3, the strong drop in the number of edges for $L \in [1, 5]$ is due to the fact that ARROW-Diff samples one random walk for every node in the graph at $L = 1$, which results in a high number of edge proposals. With increasing $L$, these edges will either be selected or dropped from the final generated graph based on the sampling on the probabilities computed by the GNN model.

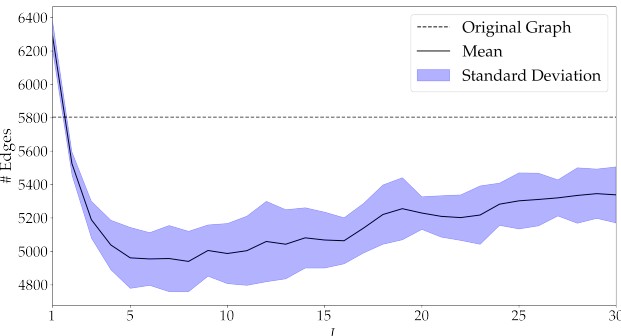

Figure 3: The mean and standard deviation of the number of edges of 10 graphs generated by ARROW-Diff, using the original node features, are reported for $L \in [1, 30]$. The dotted line represents the number of edges of the original CiteSeer graph.

## 4.3 Results and Efficiency

In Table 2, we present the results for all baselines and for ARROW-Diff trained both with and without node features on the five datasets (Table 1). The results are reported as the mean across 10 generated graphs. To evaluate the stability of all methods, we also report the standard deviation of all metrics across the 10 generated graphs in Table 4. The results presented in Table 2 indicate that ARROW-Diff achieves notable improvement across many metrics, surpassing nearly all baselines in terms of triangle counts and power-law exponent (see generated node-degree distribution in Figure 7(a)). We observe that EDGE performs the best in terms of maximum node degree across all datasets. This is due to its ability to steer the graph generation process towards a degree distribution similar to that of the original graph. We would like to highlight that the number of edges in the graphs generated by VGAE (Kipf & Welling, 2016b) and Graphite (Grover et al., 2019) was extremely high. Thus, we consider only edges for which the predicted probability $p(A_{ij} = 1|\boldsymbol{z}_i, \boldsymbol{z}_j) = \sigma(\boldsymbol{z}_i^T \boldsymbol{z}_j) > 0.95$ for nodes $i, j \in V$. Even with this constraint, the graphs generated by VGAE had over 50M edges on the Cora, DBLP, and PubMed datasets, which led to an exhaustive metric computation. Therefore, we have omitted the results for these datasets from Table 2. The scalability of ARROW-Diff is demonstrated in the substantial decrease in graph generation time (down to more than 50% compared to all baselines). This remains true even when generating very large graphs such as Cora, DBLP, and PubMed (column 'Time' in Table 2). In the presented results, the graph generation time of VGAE and Graphite is almost zero because operating on tensors of sizes up to $3,000 \times 3,000$ entries is not challenging for modern computers. The scalability of ARROW-Diff is further reflected in the training speed.

Table 2: Graph generation results of NetGAN (Bojchevski et al., 2018), VGAE (Kipf & Welling, 2016b), Graphite (Grover et al., 2019), EDGE (Chen et al., 2023), GraphRNN (You et al., 2018), BiGG (Dai et al., 2020), and ARROW-Diff with and without node features on the single, large-scale graph datasets from Table 1. The performance is given in terms of the mean of six graph statistics across 10 generated graphs. The edge overlap represents the mean overlap with the edges of the original graph. The last column reports the graph generation time for all methods, which, for ARROW-Diff, is the time for executing Algorithm 2.

| Dataset / Methods | Max. degree | Assort-ativity | Triangle Count | Power law exp. | Avg. cl. coeff. | Global cl. coeff. | Edge Overlap | Time [s] |
|---|---|---|---|---|---|---|---|---|
| *Cora-ML* | 246 | -0.077 | 5,247 | 1.77 | 0.278 | 0.004 | - | - |
| NetGAN | 181 | -0.025 | 384 | 1.67 | 0.011 | 0.001 | 3.2% | 6.2 |
| VGAE | 948 | -0.043 | 70 M | 1.66 | 0.383 | **0.002** | 22.2% | 0.0 |
| Graphite | 115 | -0.188 | 11,532 | 1.57 | **0.201** | 0.009 | 0.3% | 0.1 |
| EDGE | **202** | -0.051 | 1,410 | **1.76** | 0.064 | **0.002** | 1.3% | 5.5 |
| GraphRNN | 38 | **-0.078** | 98 | 2.05 | 0.015 | 0.042 | 0.2% | 2.0 |
| BiGG | 65 | 0.030 | 280 | 1.98 | 0.026 | 0.008 | 1.0% | 47.2 |
| ARROW-Diff | 373 | -0.112 | 5,912 | 1.81 | 0.191 | 0.001 | 57.3% | 1.8 |
| ARROW-Diff (w/o features) | 439 | -0.117 | **5,821** | 1.82 | 0.182 | 0.001 | 55.2% | 1.7 |
| *Cora* | 297 | -0.049 | 48,279 | 1.69 | 0.267 | 0.007 | - | - |
| NetGAN | 135 | 0.010 | 206 | 1.61 | 0.001 | 0.000 | 0.1% | 35.0 |
| Graphite | 879 | -0.213 | 3 M | 1.31 | **0.338** | 0.001 | 0.3% | 0.9 |
| EDGE | **248** | 0.078 | 11,196 | 1.65 | 0.021 | **0.002** | 0.2% | 85.8 |
| BiGG | 222 | -0.086 | 2,479 | 1.50 | 0.007 | 0.000 | 0.6% | 820.4 |
| ARROW-Diff | 536 | **-0.077** | 89,895 | 1.70 | 0.122 | **0.002** | 40.8% | 13.7 |
| ARROW-Diff (w/o features) | 579 | -0.080 | **77,075** | **1.68** | 0.105 | 0.001 | 42.3% | 13.4 |
| *CiteSeer* | 85 | -0.165 | 771 | 2.23 | 0.153 | 0.007 | - | - |
| NetGAN | 42 | -0.009 | 23 | 2.03 | 0.004 | 0.001 | 0.7% | 4.5 |
| VGAE | 558 | -0.036 | 15 M | 1.69 | 0.383 | 0.003 | 22.1% | 0.0 |
| Graphite | 58 | -0.198 | 2,383 | 1.70 | **0.157** | 0.016 | 0.3% | 0.1 |
| EDGE | **82** | -0.128 | 205 | 2.08 | 0.054 | 0.003 | 1.1% | 4.2 |
| GraphRNN | 31 | -0.243 | 20 | 2.88 | 0.011 | **0.007** | 0.2% | 5.7 |
| BiGG | 76 | -0.198 | 284 | 2.44 | 0.096 | 0.005 | 3.2% | 29.6 |
| ARROW-Diff | 114 | -0.192 | **795** | **2.24** | 0.109 | 0.004 | 57.8% | 1.6 |
| ARROW-Diff (w/o features) | 148 | **-0.178** | 996 | 2.15 | 0.125 | 0.003 | 63.0% | 2.0 |
| *DBLP* | 339 | -0.018 | 36,645 | 1.76 | 0.145 | 0.004 | - | - |
| NetGAN | 215 | 0.053 | 1,535 | 1.62 | 0.002 | 0.000 | 0.9% | 29.8 |
| Graphite | 734 | -0.207 | 2 M | 1.32 | 0.331 | **0.002** | 0.3% | 0.8 |
| EDGE | **258** | 0.146 | 13,423 | 1.70 | 0.018 | **0.002** | 0.4% | 62.0 |
| BiGG | 71 | 0.183 | 795 | 1.60 | 0.004 | 0.001 | 1.4% | 754.3 |
| ARROW-Diff | 478 | -0.098 | **49,865** | **1.78** | 0.069 | 0.001 | 34.2% | 11.2 |
| ARROW-Diff (w/o features) | 675 | **-0.063** | 63,017 | 1.67 | **0.079** | 0.001 | 34.9% | 9.4 |
| *PubMed* | 171 | -0.044 | 12,520 | 2.18 | 0.060 | 0.004 | - | - |
| NetGAN | **150** | **-0.021** | 184 | 1.90 | 0.001 | 0.000 | 0.1% | 39.7 |
| Graphite | 918 | -0.209 | 4 M | 1.31 | 0.341 | **0.001** | 0.3% | 1.3 |
| EDGE | 131 | 0.027 | **2,738** | **2.03** | 0.005 | **0.001** | 0.2% | 92.7 |
| BiGG | 54 | -0.183 | 5 | 2.74 | 0.000 | 0.000 | 0.0% | 265.7 |
| ARROW-Diff | 478 | -0.082 | 44,120 | 1.90 | **0.039** | 0.001 | 42.7% | 14.4 |
| ARROW-Diff (w/o features) | 474 | -0.126 | 41,379 | 1.85 | 0.034 | **0.001** | 41.2% | 12.8 |

Due to the small number of diffusion steps needed to train the random walk-based OA-ARDM (Hoogeboom et al., 2022), ARROW-Diff converges within 30 minutes on all datasets. In contrast, EDGE requires between 2-4 days on each dataset.

### 4.4 Visualization of Generated Graphs

In Figure 4, we visualize the training split of the Cora-ML and CiteSeer graphs as well as one generated graph from NetGAN, Graphite, EDGE, BiGG, and ARROW-Diff (using node features). We observe that ARROW-Diff is able to capture the basic structure of the ground truth graph and seems to generate significantly fewer edges than most of the baseline methods.

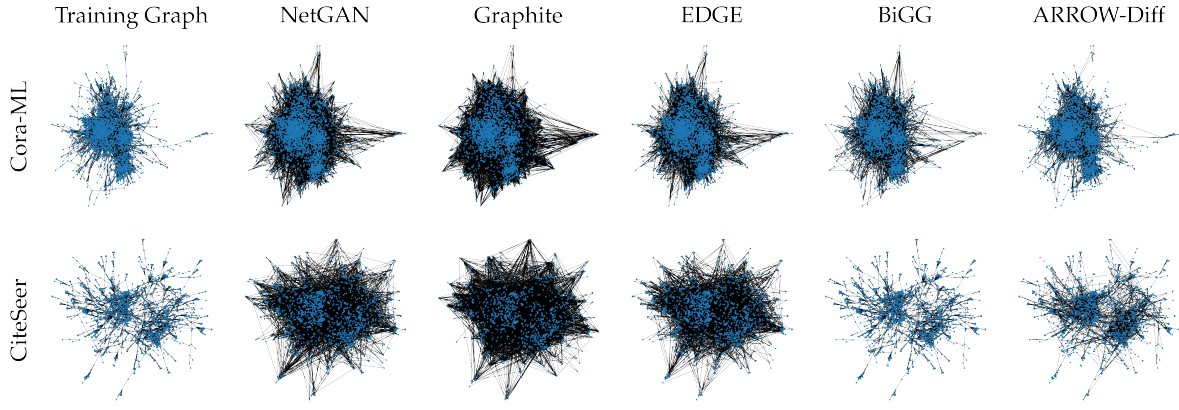

Figure 4: Visualization of the training graphs and generated graphs for Cora-ML and CiteSeer from Net-GAN, Graphite, EDGE, BiGG, and ARROW-Diff using the trained models from Table 2. We observe that ARROW-Diff is able to capture the basic structure of the original graph.

## 5 Complexity Analysis

ARROW-Diff can be used for fast generation of large graphs. In the following, we analyze the runtime complexity of ARROW-Diff, i.e., the complexity for executing Algorithm 2. Let $N$ denote the number of nodes and $|E|$ the number of edges in a graph, $D$ the random walk length which in this case is equal to the number of diffusion steps, and $L$ the number of graph generation steps (iterations) of ARROW-Diff. In each iteration $l \in [1, L]$, ARROW-Diff first samples one random walk of length $D$ for each start node $n \in V_{\text{start}}$ (line 6) to compute edge proposals. This step has a time complexity of $\mathcal{O}(ND)$ because $V_{\text{start}} \subseteq V$, and $V_{\text{start}} = V$ in the first step. Next, ARROW-Diff uses a GCN (Kipf & Welling, 2016a), to compute the probabilities for each edge in the so far generated graph including the set of newly proposed edges (line 8), which requires a maximum of $\mathcal{O}(|E|)$ operations. The computation of the new start nodes for the next iteration requires a maximum of $\mathcal{O}(|E|)$ operations with a complexity of $\mathcal{O}(|E|)$ for computing the node degrees of $\hat{G}$ (line 12), and $\mathcal{O}(N)$ to compute the probabilities (line 14) and sample the new start nodes (line 15). Overall, for $L$ generation steps, ARROW-Diff has a runtime of $\mathcal{O}(L(ND + |E|))$. This is a great reduction in complexity compared to most existing diffusion-based graph generation approaches which have a runtime complexity of $\mathcal{O}(TN^2)$ and are therefore not suitable for large-scale graph generation. The runtime of ARROW-Diff also outperforms that of EDGE (Chen et al., 2023), which is the only diffusion-based method in the literature that performs large graph generation at the magnitude we show here. We also show that ARROW-Diff has a better runtime complexity compared to the recent autoregressive diffusion-based method GraphARM (Kong et al., 2023), which has a runtime complexity of $\mathcal{O}(N^2)$, using overall $N(N-1)/2$ operations to predict the connecting edges of each node to all previously denoised nodes across all sampling steps. In Table 3 we compare the runtime complexity of the graph generation process for ARROW-Diff against that of each baseline method.

Table 3: Runtime complexity $\mathcal{O}(\cdot)$ for all baseline models and ARROW-Diff. For all methods, $N$ is the number of nodes and $M$ is the number of edges in an input graph. For NetGAN, $D$ is the random walk length, $R$ is the number of sampled random walks, and $K$ is the average degree of the nodes. For VGAE and Graphite, $F$ is the size of the latent vector. For EDGE, $K$ is the number of active nodes.

| Method | Diffusion-based | Generation Type | Runtime |
|---|---|---|---|
| NetGAN | No | Sequential | $\mathcal{O}(RD + NK)$ |
| VGAE | No | One-shot | $\mathcal{O}(FN^2)$ |
| Graphite | No | One-shot | $\mathcal{O}(FN^2)$ |
| GraphRNN | No | Sequential | $\mathcal{O}(N^2)$ |
| EDGE | Yes | Sequential | $\mathcal{O}(T \max(M, K^2))$ |
| GraphARM | Yes | Sequential | $\mathcal{O}(N^2)$ |
| BiGG | No | Sequential | $\mathcal{O}((N + M)\log(N))$ |
| ARROW-Diff | Yes | Sequential | $\mathcal{O}(L(ND + M))$ |

## 6 Limitations

While ARROW-Diff demonstrates promising results for large-scale graph generation, several limitations must be considered. First, the runtime complexity for inference of the GNN component ($\mathcal{O}(|E|)$) with increasing graph size might present a bottleneck, limiting the efficiency and scalability of ARROW-Diff to very large graphs. Another limitation of ARROW-Diff is that it generates graphs with the same number of nodes as the original graph, due to the OA-ARDM framework (Hoogeboom et al., 2022). Addressing these limitations in future work could provide a more comprehensive evaluation of ARROW-Diff's performance and scalability across graphs with various sparsity and configurations.

Potential future work could focus on an adaptation of ARROW-Diff for learning on multiple graphs which would enable the evaluation of ARROW-Diff on a wider range of graph structures. Future experiments should also incorporate more comprehensive hyper-parameter tuning for the reported baselines. Relying on default hyper-parameters for each method may potentially undermine their performance. Finally, investigating the effect of the random walk length on the results could provide valuable insights into ARROW-Diff's sensitivity to this parameter.

## 7 Conclusion

In this paper, we present ARROW-Diff, a novel approach for large-scale graph generation based on random walk diffusion. ARROW-Diff generates graphs by integrating two components into an iterative procedure: (1) An order agnostic autoregressive diffusion model on the level of random walks that learns the generative process of random walks of an input graph, and (2) a GNN component that learns to filter out unlikely edges from the generated graph. Due to the random walk-based diffusion, ARROW-Diff efficiently scales to large graphs, significantly reducing the generation time compared to baselines. In our experiments, we show how ARROW-Diff generates graphs of high quality, reflected in the high performance on many graph statistics.

## Acknowledgements

Tobias Bernecker and Ghalia Rehawi are supported by the Helmholtz Association under the joint research school "Munich School for Data Science - MUDS". Francesco Paolo Casale was funded by the Free State of Bavaria's Hightech Agenda through the Institute of AI for Health (AIH). Janine Knauer-Arloth's contributions were supported by the Brain & Behavior Research Foundation (NARSAD Young Investigator Grant, #28063). Annalisa Marsico acknowledges support by the BMBF Cluster4Future program CNATM.

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

# A    Standard Deviations of the Results in Table 2

Table 4: Standard deviation for each metric shown in Table 2.

| *Dataset* / Methods | Max. degree | Assort-ativity | Triangle Count | Power law exp. | Avg. cl. coeff. | Global cl. coeff. | Edge Overlap | Time [s] |
|---|---|---|---|---|---|---|---|---|
| *Cora-ML* | 246 | -0.077 | 5,247 | 1.77 | 0.278 | 0.004 | - | - |
| NetGAN | 11 | 0.004 | 19 | 0.00 | 0.001 | 0.000 | 0.2% | 1.0 |
| VGAE | 24 | 0.003 | 2 M | 0.11 | 0.001 | 0.000 | 0.6% | 0.0 |
| Graphite | 14 | 0.015 | 1,411 | 0.02 | 0.007 | 0.001 | 0.1% | 0.0 |
| EDGE | 1 | 0.010 | 125 | 0.00 | 0.003 | 0.000 | 0.1% | 3.7 |
| GraphRNN | 20 | 0.085 | 38.558 | 0.210 | 0.004 | 0.048 | 0.09% | 1.47 |
| BiGG | 45 | 0.029 | 65 | 0.02 | 0.004 | 0.003 | 0.1% | 1.8 |
| ARROW-Diff | 9 | 0.002 | 320 | 0.02 | 0.007 | 0.000 | 1.5% | 0.0 |
| ARROW-Diff (w/o features) | 13 | 0.004 | 326 | 0.01 | 0.006 | 0.000 | 1.5% | 0.0 |
| *Cora* | 297 | -0.049 | 48,279 | 1.69 | 0.267 | 0.007 | - | - |
| NetGAN | 13 | 0.003 | 13 | 0.00 | 0.000 | 0.000 | 0.0% | 0.9 |
| Graphite | 79 | 0.005 | 116,224 | 0.00 | 0.004 | 0.000 | 0.0% | 0.0 |
| EDGE | 3 | 0.016 | 1,111 | 0.00 | 0.001 | 0.000 | 0.0% | 0.8 |
| BiGG | 73 | 0.028 | 1,042 | 0.05 | 0.001 | 0.000 | 0.1% | 127.9 |
| ARROW-Diff | 15 | 0.003 | 2,040 | 0.00 | 0.002 | 0.000 | 0.7% | 0.7 |
| ARROW-Diff (w/o features) | 35 | 0.002 | 4,714 | 0.01 | 0.003 | 0.000 | 1.3% | 1.3 |
| *CiteSeer* | 85 | -0.165 | 771 | 2.23 | 0.153 | 0.007 | - | - |
| NetGAN | 8 | 0.019 | 6 | 0.01 | 0.002 | 0.000 | 0.2% | 0.2 |
| VGAE | 16 | 0.004 | 337,244 | 0.11 | 0.001 | 0.000 | 0.8% | 0.1 |
| Graphite | 6 | 0.022 | 419 | 0.02 | 0.010 | 0.001 | 0.1% | 0.0 |
| EDGE | 0 | 0.011 | 27 | 0.01 | 0.007 | 0.000 | 0.2% | 3.3 |
| GraphRNN | 8 | 0.054 | 5.76 | 0.367 | 0.003 | 0.004 | 0.07% | 0.6 |
| BiGG | 0 | 0.000 | 1 | 0.00 | 0.001 | 0.000 | 0.1% | 1.1 |
| ARROW-Diff | 5 | 0.007 | 69 | 0.03 | 0.008 | 0.000 | 1.4% | 0.0 |
| ARROW-Diff (w/o features) | 8 | 0.008 | 68 | 0.02 | 0.008 | 0.000 | 1.1% | 0.2 |
| *DBLP* | 339 | -0.018 | 36,645 | 1.76 | 0.145 | 0.004 | - | - |
| NetGAN | 10 | 0.005 | 69 | 0.00 | 0.000 | 0.000 | 0.0% | 0.4 |
| Graphite | 74 | 0.004 | 69,026 | 0.00 | 0.003 | 0.000 | 0.0% | 0.0 |
| EDGE | 2 | 0.033 | 1,675 | 0.00 | 0.002 | 0.000 | 0.0% | 0.5 |
| BiGG | 12 | 0.022 | 57 | 0.01 | 0.000 | 0.000 | 0.1% | 11.7 |
| ARROW-Diff | 26 | 0.002 | 2,202 | 0.01 | 0.002 | 0.000 | 0.8% | 0.7 |
| ARROW-Diff (w/o features) | 27 | 0.003 | 2,797 | 0.01 | 0.002 | 0.000 | 0.6% | 0.0 |
| *PubMed* | 171 | -0.044 | 12,520 | 2.18 | 0.060 | 0.004 | - | - |
| NetGAN | 14 | 0.004 | 9 | 0.00 | 0.000 | 0.000 | 0.0% | 1.8 |
| Graphite | 70 | 0.005 | 244,816 | 0.00 | 0.004 | 0.000 | 0.0% | 0.0 |
| EDGE | 4 | 0.038 | 741 | 0.00 | 0.001 | 0.000 | 0.0% | 0.5 |
| BiGG | 7 | 0.010 | 3 | 0.16 | 0.000 | 0.000 | 0.0% | 32.5 |
| ARROW-Diff | 11 | 0.003 | 1,454 | 0.00 | 0.001 | 0.000 | 0.8% | 1.1 |
| ARROW-Diff (w/o features) | 15 | 0.004 | 2,427 | 0.01 | 0.001 | 0.000 | 1.0% | 0.0 |

# B    Evaluation of the Quality of the Generated Graphs

In the following analysis, we validate the capability of ARROW-Diff in learning the topological structure of the original graph and in generating new graphs that have a similar data distribution as the original graph. To this end, we generated 100 graphs for each of CiteSeer, DBLP, and PubMed graphs. We used the graph embedding technique Graph2Vec (Narayanan et al., 2017) to map the generated graphs into embeddings of size 256. A principal component analysis (PCA) of the embeddings (Figure 5) revealed a clear separation between the different generated graphs of the different datasets highlighting the ability of ARROW-Diff in generating graphs with similar structural properties for the same dataset. The results also depicted that the embeddings of the generated graphs are close in the embedding space to their original counterparts, indicating that ARROW-Diff produces graphs that closely mimic the structural properties of the input graphs and underscoring the fidelity of the generated graphs. A logistic regression-based classifier trained on the generated graphs and tested on the real graphs reported a Classification Accuracy Score (CAS) (Ravuri & Vinyals, 2019) of 0.67.

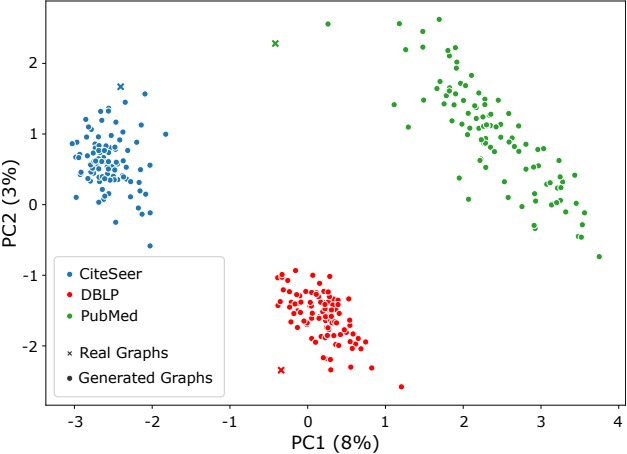

Figure 5: Principal component analysis (PCA) of the Graph2Vec embeddings of the original CiteSeer, DBLP, and PubMed graphs and 100 generated graphs for each of the datasets, showing a clear separation between the generated graphs of the different datasets, and a proximity of the generated graphs to their real graph counterparts.

# C    Supplementary Experiments and Results

## C.1    Stochastic Block Model Dataset

In the following experiment, we evaluate the ability of ARROW-Diff to generate graphs with a different structure to those shown in Section 4. Specifically, we create a synthetic dataset containing a Stochastic Block Model (SBM) (Holland et al., 1983) graph that does not show a preferential attachment property (power-law degree distribution). The constructed SBM graph includes three communities with 60, 100, and 200 nodes, and a total of 3,824 edges. Similar to our main experiments (Section 4), we split the edges into training, validation, and test edges, and use the same training split across all experiments.

## C.2    Parameters for ARROW-Diff Training and Graph Generation

Due to the smaller size of the SBM graph, we set the random walk length to $D = 8$ and run ARROW-Diff for $L = 5$ iterations only. We also set the dimensionality of positional encodings (used by the GNN component) of the nodes to 64.

Table 5: The results of ARROW-Diff and six baseline models on the SBM dataset with 360 nodes and 3,824 edges.

| Dataset
Methods | Max.
degree | Assort-
ativity | Triangle
Count | Avg.
cl. coeff. | Global
cl. coeff. | Edge
Overlap | Time
[s] |
|---|---|---|---|---|---|---|---|
| SBM | 36 | -0.016 | 3,623 | 0.134 | 0.019 | - | - |
| NetGAN | 46 | **-0.013** | 2,393 | 0.080 | 0.010 | 9.2% | 1.5 |
| VGAE | 122 | -0.012 | 146,304 | 0.383 | 0.013 | 22.1% | 0.0 |
| Graphite | 11 | -0.134 | 23 | 0.048 | 0.082 | 0.3% | 0.1 |
| EDGE | 34 | 0.017 | 1,581 | 0.072 | 0.012 | 5.7% | 1.6 |
| GraphRNN | 14 | -0.061 | 57 | 0.025 | 0.038 | 1.7% | 0.2 |
| BiGG | **33** | -0.029 | 2,204 | **0.113** | **0.019** | 65.9% | 13.9 |
| ARROW-
Diff (w/o features) | 48 | 0.026 | **2,969** | 0.157 | 0.020 | 40.5% | 0.5 |

## C.3 Results

Table 5 presents the results as the mean across 10 generated graphs. For evaluation, we omit the power-law exponent metric, as it is exclusively applicable to scale-free graphs. The results demonstrate that ARROW-Diff achieves the best performance on the triangle count metric, consistent with the results on the citation graphs (Table 2). It also shows competitive performance compared to other baselines on the rest of the metrics, performing second best on two metrics while significantly reducing graph generation time. In Figure 6, we provide a visualization of the training split of the synthetic SBM graph as well as the generated graphs, showcasing the ability of ARROW-Diff in capturing the three community structures of the SBM graph.

Training Graph      NetGAN      Graphite      EDGE      BiGG      ARROW-Diff

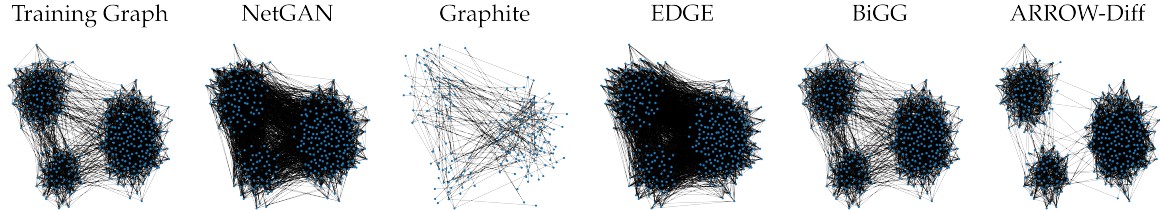

Figure 6: Visualization of the SBM training graph as well as generated graphs from NetGAN, Graphite, EDGE, BiGG, and ARROW-Diff. We observe that ARROW-Diff is able to capture the community structures of the original graph.

# D    Node Degree Distribution of Generated Graphs

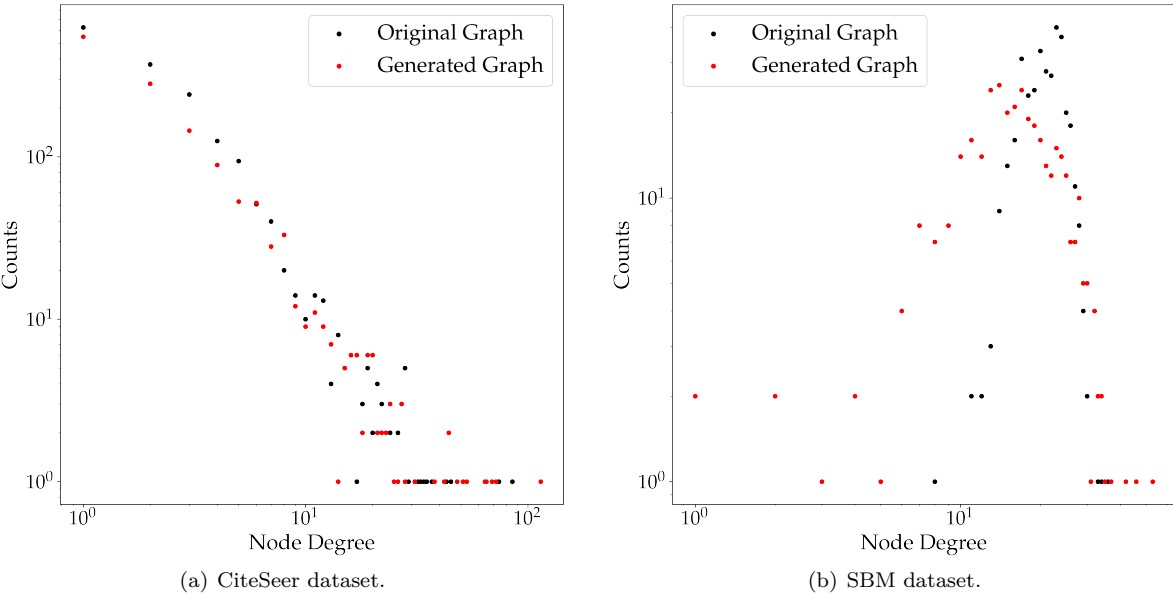

(a) CiteSeer dataset.

(b) SBM dataset.

Figure 7: Node degree distributions of the real CiteSeer and SBM graphs and one generated graph of ARROW-Diff for each of the two datasets. The original node features are used by ARROW-Diff for the CiteSeer dataset.

