# OpenReview forum: "Random Walk Diffusion for Efficient Large-Scale Graph Generation"
_TMLR — Accepted by TMLR_

### Review · Reviewer_xGqN · 2024-11-10

**Summary Of Contributions:**

This paper proposes ARROWDiff a methods that learns to generate large graphs of a similar structure as a sample original graph.
ARROWDiff uses a learned diffusion model to generate random walks from the original graph based on previous work by Hoogeboom et al. (2022), first sampling random walks and then masking nodes in forward diffusion to predict nodes in reverse diffusion. It also uses a GNN to capture the topology of the original graph and thereby assess the validity of edges proposed by the diffusion model.
The components work together in an iterative process to generate the final graph.

**Audience:**

Yes

**Broader Impact Concerns:**

No concerns.

**Claims And Evidence:**

Yes

**Requested Changes:**

The proposed method works reasonably well, in comparison to alternative graph generation models.
However, the comparison seems to be unfair, since the proposed method draws heavily from the graph embedding literature, yet it is not assesed in comparison to that literature. Graph embedding methods like DeepWalk, node2vec, VERSE, and NetSMF, can be used directly for reconstructing an embedded graph, which amounts in practice to a graph generation process in disguise. Such a usage of a graph embedding, which is practices in the embedding literature, comes very close to what is proposed in this work. Therefore, the study should assess the proposed method vs. graph embedding methods employed for generating new graphs similar to the embedded graph.

**Strengths And Weaknesses:**

Strength:
The work proposes a novel method for graph generation, combining a diffusion model and a GNN.
The proposed model, ARROWDiff, outperforms previous graph generation methods in terms of fidelity to original graph characteristics.

Weakness:
Nevertheless, the experimental results show that, even within 30 iteration, the generated graph do not approach the original graphs interms of several characteristics, especially globale clustering coefficient, while they deviate significantly in terms of maximum degree.

---

> ### Author Response · Authors · 2025-01-10
>
> We would like to thank the reviewer for their constructive feedback.
>
> ### Answer regarding the mentioned ARROW-Diff's weakness:
>
> The reviewer highlights a key challenge in graph generation: Achieving accurate reproduction of various graph characteristics. While ARROW-Diff demonstrates strong performance in capturing specific properties like the number of triangles and power law exponent, we acknowledge that certain metrics, notably the global clustering coefficient, and maximum degree, still exhibit a gap between generated and original graphs. This observation aligns with the broader limitations of current graph generation techniques, where perfectly replicating all graph characteristics remains an open research area. Our primary contribution with ARROW-Diff is the scalability and efficiency of generating very large graphs, which covers a gap in the current state of the art. We hope future work will focus on enhancing ARROW-Diff's ability to capture a wider range of graph properties, including those highlighted by the reviewer.
>
> ### Answer regarding the requested changes:
>
> We acknowledge the reviewer's point that a comparison to graph embedding methods used for generation is essential. In our experiments, we compare with methods that are designed for generating new graphs and which use node-level embeddings learned via graph convolutional neural networks, e.g. Graphite and VGAE. We also compare with NetGAN, a random walk-based method for graph generation using GANs, which is the most similar method in terms of methodology to our approach. Specifically, NetGAN and ARROW-Diff both use edges of generated random walks as possible edges (edge proposals) for the generation of the final graph.
>
> While our method draws inspiration from graph embedding techniques (basically the idea that graph structure can be captured by random walks), there are some distinctions to the mentioned methods to be made:
>
> - Primary goal: Methods like DeepWalk, node2vec, VERSE, and NetSMF aim to learn low-dimensional node representations, best suited for downstream tasks such as node classification and link prediction. In contrast, ARROW-Diff's primary goal is directly generating new graph structures that exhibit similar properties to the input graph.
>
> - Graph reconstruction vs. generation: The graph reconstruction process based on learned embeddings mentioned in VERSE leads to the same reconstructed graph every time. Hence, there is no variability when sampling many new graphs. It would require additional steps to convert learned embeddings back into graph structures while at the same time introducing variability to newly created graphs, something that is not covered by these methods. ARROW-Diff, by design, aims to sample from the learned data distribution of the input graph, enabling the generation of diverse and novel graph instances.
>
> We acknowledge that these methods, originally designed for node representation learning, can be adapted for graph generation. However, a thorough comparison with these adapted methods would require further investigation and is left for future work.

---

> > ### Comment · Reviewer_xGqN · 2025-02-10
> > **Cannot recommend acceptance when authors refuse an essential comparison**
> >
> > The authors do not wish to perform the requested comparison to graph embedding methods employed for generating new graphs similar to the embedded graph, even while they acknowledge that these methods can be adapted for graph generation and that such a comparison is essential. That leaves me in a position where I cannot recommend acceptance.

---

> > > ### Author Response · Authors · 2025-02-10
> > >
> > > We thank the reviewer for their comments. The reviewer suggested a comparison of our method to graph embedding methods employed for generating new graphs similar to the embedded graph. Indeed, in our experiments, we compare ARROW-Diff to six different baselines including two embeddings-based techniques that use node embeddings to generate new graphs, namely VGAE and Graphite. Similar to the methods suggested by the reviewer like DeepWalk and node2vec, VGAE and Graphite use an embedding technique, specifically node embeddings extracted from a GCN, to generate new graphs from the learned embeddings. Therefore we believe that the reviewer’s point is covered in our experiments. We apologize if our point was poorly communicated in our initial reply. If the reviewer thinks that their concern is not addressed, we would appreciate the reviewer’s explanation of why a comparison with VGAE and Graphite is not sufficient to cover this point.

---

> > > > ### Comment · Reviewer_xGqN · 2025-02-10
> > > > **Substitution is not comparison**
> > > >
> > > > The comparison done is insufficient because using A in place of B does not imply a comparison to B.

---

### Review · Reviewer_iqjV · 2024-11-25

**Summary Of Contributions:**

The paper  introduces ARROW-Diff (AutoRegressive RandOm Walk Diffusion), a novel approach for generating large-scale graphs efficiently. The method addresses the limitations of previous diffusion-based graph generation techniques, which often struggle with scalability and quality when applied to large graphs. ARROW-Diff employs a two-component system:

1. Random Walk-Based Diffusion Model: This component uses an order-agnostic autoregressive diffusion framework (OA-ARDM) to sample random walks from an input graph. The random walk-based diffusion captures the complex and sparse structure of graphs, allowing ARROW-Diff to scale efficiently.

2. Graph Neural Network: This component predicts the validity of edges in the generated random walks, refining the graph generation process and ensuring high-quality outputs.

ARROW-Diff iteratively integrates these components, guided by node degrees, to produce graphs with similar degree distributions to the original graph. The method demonstrates superior performance in terms of generation time and graph quality compared to existing baseline methods, efficiently handling graphs with up to 20,000 nodes.

**Audience:**

Yes

**Broader Impact Concerns:**

Not applicable.

**Claims And Evidence:**

Yes

**Requested Changes:**

Some sections could benefit from additional clarity, particularly in explaining complex concepts like OA-ARDM and its integration with GNNs for readers unfamiliar with these models.

While the paper addresses many challenges in large-scale graph generation, it would be beneficial to discuss potential limitations of ARROW-Diff, such as its applicability to different types of graphs or potential issues with very sparse or dense graphs. Suggestions for future work could include exploring enhancements in the GNN component or adapting the method for dynamic graph scenarios.

**Strengths And Weaknesses:**

Strengths:

The paper presents a significant advancement in the field of graph generation by introducing a scalable method that leverages random walks for diffusion processes. This approach is innovative as it reduces computational complexity and overcomes limitations related to edge independence and scalability found in existing methods. The methodology is well-articulated, with clear descriptions of both components—random walk-based diffusion and GNN validation. The use of OA-ARDM for random walk sampling is particularly noteworthy as it allows for efficient scaling by limiting diffusion steps to the length of the random walk rather than the size of the entire graph. The experiments conducted demonstrate that ARROW-Diff outperforms other methods in terms of both speed and quality of generated graphs. The results are compelling, showing significant reductions in generation time while maintaining or improving graph statistics such as degree distribution.

Overall, this paper makes a substantial contribution to graph generation research by introducing a novel method that effectively balances efficiency and quality. It is recommended for acceptance after minor revisions to enhance clarity and address any potential limitations in broader applications.

Weaknesses:

Scalability Concerns: Although ARROW-Diff is designed to handle large graphs, its performance may still be limited by the complexity of the Graph Neural Network (GNN) component. As the size of the graph increases, the computational cost associated with training and inference using GNNs can become significant, potentially offsetting the efficiency gains from the random walk-based diffusion model.

Graph Quality Variability: The quality of the generated graphs, particularly in terms of preserving complex topological features like community structures or clustering coefficients, may vary. The method's reliance on random walks might not fully capture intricate dependencies present in certain types of graphs, leading to discrepancies between generated and original graph statistics.

Edge Validity Prediction: The GNN component's ability to predict edge validity accurately is crucial for ensuring high-quality graph generation. However, if the GNN model is not well-tuned or if the training data is not representative of the target graph's structure, this can lead to errors in edge prediction, affecting the overall quality of the generated graph.

Dependency on Node Degree Guidance: The iterative process guided by node degrees could potentially introduce biases in graph generation. If the degree distribution of the input graph is not well-represented or if there are anomalies, it might lead to inaccurate reproduction of graph properties in the generated output.

Limited Generalizability: The method's effectiveness across different types of graphs (e.g., social networks vs. molecular structures) might be limited. The specific design choices, such as random walk length and diffusion steps, may need significant adjustments to cater to diverse graph structures and sizes.

---

> ### Author Response · Authors · 2025-01-15
>
> We would like to thank the reviewer for their valuable feedback, we discuss the reviewer's concerns in the following points :
> - Scalability Concerns: We agree with the reviewer's assessment. For graphs with the magnitude tested in our work (up to ~20k nodes, and ~60k edges), the training and prediction of the GNN component is efficient. However, more investigation needs to be done for even larger graphs. We discuss this in a newly added Section 6 ‘Limitations’.
>
> - Edge Validity Prediction: The reviewer correctly points out the critical role of accurate edge validity prediction by the GNN component. If the GNN misclassifies edges, it can lead to the generation of unrealistic or low-quality graphs. To reduce this risk, we employ several strategies: 1) We use a perturbed version (training graph) of the original graph to train the GNN for edge classification/validation.  2) We use a validation split and early stopping on the validation loss to ensure adequate training. We have updated the paper (Section 4.1 in paragraph 'ARROW-Diff Models Training') to clarify this.
>
> - Dependency on Node Degree Guidance: We would like to thank the reviewer for raising this point. ARROW-Diff generates graphs with a similar node-degree distribution to the real graphs. This is reflected in the metric ‘power-law exponent’ in Table 2, which is calculated by estimating the power-law exponent of the generated and real node degree distributions and comparing them, if they are close then this suggests similar distributions. Figure 6 shows an example of node degree distributions of a real and a generated graph. The result shows a similar node degree distribution between the two.
>
> - Graph quality variability (Generalizability): We thank the reviewer for raising this critical point. We acknowledge the importance of evaluating ARROW-Diff's performance in capturing different graph structures besides the ones tested in the current experiments. Hence, we have added a new experiment (Appendix Section C and D), in which we test the performance of ARROW-Diff on a synthesized Stochastic Block Model (SBM) dataset with three communities (360 nodes and 3824 edges), which also has higher edge density than the Citation graphs.
>
> ### Regarding requested changes:
> - We acknowledge that some sections, particularly those explaining OA-ARDM and its integration with GNNs, could benefit from more detailed explanations for readers unfamiliar with these models. Due to page limit constraints, we were unable to extend too much on the existing text. Still, we have updated the text to make the integration of the two components of ARROW-Diff clearer (Section 3 of the revised manuscript).
>
> - In the revised manuscript, we have added the section ‘Limitations’ (Section 6) in which we discuss several limitations of ARROW-Diff, such as limitations regarding the GNN component and adapting ARROW-Diff to facilitate training on multiple graphs and hence evaluating a wider range of graph topologies.

---

### Review · Reviewer_TkfS · 2024-12-13

**Summary Of Contributions:**

The paper presents ARROW-Diff, a graph generative model targeting in particular large graphs, constructed by combining OA-ARDMs (order agnostic autoregressive diffusion models, a variant trained with training time order-agnostic masking to train without reliance on order) with the random walk graph generation pioneered by netgan and node2vec, as well as the concept of active nodes and degree guidance pioneered by EDGE, using a GNN to commit to edges identified in the random walk as candidates..

The paper is evaluated against NetGAN, Graphite,EDGE on large citation graph datasets, as well as against VGAE and GraphRNN on the smallest 2., showing strong results

**Audience:**

Yes

**Broader Impact Concerns:**

Aside from the impending climate catastrophy which we all contribute to blasting GPUs, I think the authors are in the clear.

**Claims And Evidence:**

Yes

**Requested Changes:**

Critical:
- acknowledge or rebut the limitations I pointed out, add the missing related works or justify their omission
- mention the baselines, if computationally feasible, compare against them
- ensure that the results are statistically meaningful (be it via boostraps, CAS or both)
- add at least one real world adversarial dataset (not exhibition preferential attachment), or even better, synthetic benchmarks to test corners of the network

Bonus/would strengthen:
- all the computationally expensive extra work I pointed out.

**Strengths And Weaknesses:**

Strengths:

1. The paper reports strong results, in particular in edge overlap and metrics like assortativity, while benefiting form improved scaling via relatively short random walks chopping the graph up efficiently into parallélisable chunks

2. The idea is clever, it inherits the BA like preferential attachment pattern from netgan/ node2vec but but can do multi-step candidate selection as well as correction via the EDGE inspired guidance GNN


Weaknesses:

- mainly in the evaluation/presentation:

    - the authors use the paper hyperparameters, but some some of these papers are years old. I myself have made this mistake of accidentally sandbagging papers. For a fair comparison, one would need to estimate total numbers of flops and parameters and spend equal resources tuning. This is going beyond what is common in ML and computationally expensive, however I think *acknowledging this* not only in the end of the paper, but also in the experimental setup is important.
   - in particular, the authors match the random walk length method to the dataset characteristics, which is a level of tuning not all the baselines performed
    - While I think the authors are in the clear, I would recommend using a form of bootstrap (chopping generated and original graphs into equally sized random overlapping communities computing metrics on these, then aggregating) to construct confidence intervals and perform p-tests for statistically significant results (Efron has a nice book introducing bias corrected bootstrapping methods). Alternatively one could use https://arxiv.org/abs/1905.10887 with the generated graphs as training set to evaluate on the real graph

    - Multiple random seeds should also ideally be taken for _training_, or acknowledgement of such shortcoming should be made

    - There should be a standardizd callibration procedure for the edge probability selection

    - there are some important strong baselines omitted:

        - [https://bmcbioinformatics.biomedcentral.com/articles/10.1186/1471-2105-11-213](https://bmcbioinformatics.biomedcentral.com/articles/10.1186/1471-2105-11-213) an autoregressive model which can scale to > 100k nodes

        - [https://openreview.net/forum?id=2XkTz7gdpc](https://openreview.net/forum?id=2XkTz7gdpc) on a high level “EDGE, but spectrally guided”

        - [https://arxiv.org/abs/2311.02142](https://arxiv.org/abs/2311.02142) “Digress, but learning degree guidance”

    - there is also a nice chance to discuss whether the permutation-augmentation technique of OA-ARDMs will work for graphs, [https://arxiv.org/abs/2307.01646v4](https://arxiv.org/abs/2307.01646v4) is a work that argues yes, and [https://arxiv.org/abs/2110.02096](https://arxiv.org/abs/2110.02096) showed that *any* algorithm with a permutation invariant *loss* can work, however, this will only hold if one can actually *train* sufficiently. on the given tasks, I think this is a given, but worth discussion

    - while the baselines are inadvertently sandbagged (again, not a diss, I’ve done this myself by accident), the datasets are inadvertently cherry picked: they all exhibit BA style preferential attachment properties, which is playing to this algorithms strength. I’d like to see at least synthetic baselines on more structured graphs, maybe pointclouds/ grids, etc. again, not a grounds for reject IFF it is acknowledged as a limit of the evaluation

---

> ### Author Response · Authors · 2025-01-15
>
> We would like to thank the reviewer for their valuable and constructive feedback. We address the reviewer's concerns in the following points:
>
> - The use of default hyperparameters for the baseline methods: This is indeed an important point. We discuss this as a limitation in the newly added Section 6 (Limitations).
>
> - Matching the random walk length method to the dataset characteristics: The random walk length is an integral aspect of our method's design. Indeed it is important to recognize that fine-tuning this parameter plays a crucial role in the effectiveness of our method. We would like to highlight that we tested the performance of ARROW-Diff using multiple walk lengths and 16 was the best-performing value. Therefore, across all datasets, we set the random walk length to 16. This is similar to the work of Bojchevski et al. (NetGAN) in which the authors performed a hyperparameter tuning on the random walk length.
> In the newly added Section 6 (Limitations), however, we discuss the importance of tuning and investigating the effect of this parameter on the performance of ARROW-Diff.
>
> - Statistical significance of the results: We appreciate the reviewer's valuable feedback and acknowledge the importance of the suggested analysis. We have followed the suggested Classification Accuracy Score (CAS) approach and added the results of this analysis to the revised manuscript (Supplementary Section B):
>
>   "We validated the capability of ARROW-Diff in learning the topological structure of the original graph and in generating new graphs that
>   have similar data distribution as the original graph. To this end, we generated 100 graphs for each of CiteSeer, DBLP, and PubMed
>   graphs. We used the graph embedding technique Graph2Vec to map the generated graphs into embeddings of size 256. A principal
>   component analysis (PCA) of the embeddings (Figure 5) revealed a clear separation between the different generated graphs of the
>   different datasets highlighting the ability of ARROW-Diff to generate graphs with similar structural properties for the same dataset. The
>   results also depicted that the embeddings of the real graphs are close in the embedding space to their generated counterparts,
>   indicating that ARROW-Diff produces graphs that closely mimic the structural properties of the input graphs and underscoring the
>   fidelity of the generated graphs. A logistic regression-based classifier trained on the generated graphs and tested on the real graphs
>   gives a Classification Accuracy Score (CAS) of 0.67."

---

> ### Author Response · Authors · 2025-01-15
>
> - The use of multiple seeds: We agree with the reviewer that ideally one should train on multiple seeds. We are accounting here for variability by rather generating graphs multiple times and averaging our metrics for different generated graphs.
>
> - Calibration process for edge probability selection: The reviewer raises a valid point regarding the need for a standardized calibration procedure for edge probability selection. While we acknowledge the potential benefits of calibration, we have chosen not to implement it in the current version of ARROW-Diff. This is because adding a calibration module changes the output of the GNN and would require that we rerun our method on all datasets. However, we recognize the potential benefits of calibration and believe it is an interesting direction for future research.
>
> - Additional baselines: We thank the reviewer for bringing these methods to our attention. Indeed we found that it is important to compare with the [BiGG](https://arxiv.org/pdf/2006.15502) method which is a recent autoregressive-based graph generation approach that reports reduced generation time and scalability to large graphs. We have added BiGG as another baseline in our experiments and updated the manuscript accordingly.
> We choose not to include the second paper “Efficient and Scalable Graph Generation through Iterative Local Expansion”, a one-shot method using denoising diffusion since we compare against two other methods that use the same generation strategy (one-shot) which are VGAE and Graphite. Moreover, this method is reported to scale to graphs up to 5k nodes only. We also choose not to include the third paper “Sparse Training of Discrete Diffusion Models for Graph Generation” since this method reports scaling to graphs with up to 500 nodes only.
>
> - Discussing permutation-augmentation technique of OA-ARDMs for graphs: We thank the reviewer for pointing out this interesting topic. We would like to refer here to the work of Kong et al. (GraphARM) (which we mention in our paper). In this work, the authors apply an autoregressive diffusion model directly to the graph structure. The authors argue that “While ARDM imposes a uniform ordering for arriving at an order agnostic variational lower bound (VLB) of likelihood, a random ordering fails to capture graph topology”, which they investigate in their Appendix A.1. Hence they introduce a node-ordering generated by a diffusion ordering network.
> However, in our case, we train an OA-ARDM to generate random walks. Therefore, the generation order of the OA-ARDM has no influence on the permutation order of the adjacency matrix of the generated graph. Instead, the generated random walks are used as edge proposals, which are later filtered by a GNN.
>
> - Showing the performance of ARROW-Diff on a non-scale-free dataset:  We thank the reviewer for raising this critical point. We acknowledge the importance of evaluating ARROW-Diff's performance in capturing different graph structures besides the ones tested in the current experiments. Hence, we have added a new experiment (Appendix Section C and D), in which we test the performance of ARROW-Diff on a synthesized Stochastic Block Model (SBM) dataset with three communities (360 nodes and 3824 edges), which also has higher edge density than the Citation graphs.

---

### Author Response · Authors · 2025-01-17
**Summary for all reviewers**

We would like to thank all reviewers for their effort and valuable comments and suggestions.
In the following, we summarize all additional analyses we added to the revised manuscript.

1- A new baseline BiGG (https://arxiv.org/pdf/2006.15502), which is an autoregressive graph generative approach: Table 2 of the revised manuscript.

2- An additional synthetic dataset (Stochastic Block Model SBM) to evaluate the ability of ARROW-Diff to generate more structured graphs different from the scale-free topology graphs tested in the original experiments: Appendix Section C (Supplementary Experiments and Results).

3- A statistical significance analysis of the results using the Clustering Accuracy Score (CAS): Appendix Section B.

4- A new limitation section to discuss all limitations mentioned by the reviewers: Section 6.

5- Reformulations to clarify some details regarding the integration of the OA-ARDM with the GNN component: Section 3 (Methods) of the revised manuscript.

A new revised manuscript and code (supplementary material) are now uploaded.

---

### Decision · Action_Editor_13VJ · 2025-02-12

**Recommendation:** Accept as is

**Comment:**

This paper proposes a generative model for large graphs. Reviewers appreciated the proposed approach and the paper is well-written. I believe the paper meets TMLR's acceptance criteria and thus recommend acceptance.

**Audience:**

Yes.

**Claims And Evidence:**

Two reviewers agree the paper presents enough evidence to back up its claims, whereas another reviewer complains about comparisons against missing baselines. The latter reviewer did not engage with the authors during the rebuttal to better explain their rationale, and I side with the other two reviewers.